# Bonding Performance of Glass Fiber-Reinforced Polymer Bars under the Influence of Deformation Characteristics

**DOI:** 10.3390/polym15122604

**Published:** 2023-06-07

**Authors:** Fang Xie, Wanming Tian, Pedro Diez, Sergio Zlotnik, Alberto Garcia Gonzalez

**Affiliations:** 1Department of Civil Engineering, Shaoxing University, Shaoxing 312000, China; tianwanmingsx@163.com; 2Department of Civil and Environmental Engineering, Universitat Politècnica de Catalunya Barcelona Tech, 08034 Barcelona, Spain; pedro.diez@upc.edu (P.D.); sergio.zlotnik@upc.edu (S.Z.); berto.garcia@upc.edu (A.G.G.)

**Keywords:** GFRP bar, bond–slip behavior, deformation coefficient, engineering performance, four-fold model

## Abstract

Glass fiber-reinforced polymer (GFRP) of high performance, as a relatively ideal partial or complete substitute for steel, could increase the possibility of adapting structures to changes in harsh weather environments. While GFRP is combined with concrete in the form of bars, the mechanical characteristics of GFRP cause the bonding behavior to differ significantly from that of steel-reinforced members. In this paper, a central pull-out test was applied, according to ACI440.3R-04, to analyze the influence of the deformation characteristics of GFRP bars on bonding failure. The bond–slip curves of the GFRP bars with different deformation coefficients exhibited distinct four-stage processes. Increasing the deformation coefficient of the GFRP bars is able to significantly improve the bond strength between the GFRP bars and the concrete. However, while both the deformation coefficient and concrete strength of the GFRP bars were increased, the bond failure mode of the composite member was more likely to be changed from ductile to brittle. The results show members with larger deformation coefficients and moderate concrete grades, which generally have excellent mechanical and engineering properties. By comparing with the existing bond and slip constitutive models, it was found that the proposed curve prediction model was able to well match the engineering performance of GFRP bars with different deformation coefficients. Meanwhile, due to its high practicality, a four-fold model characterizing representative stress for the bond–slip behavior was recommended in order to predict the performance of the GFRP bars.

## 1. Introduction

The global climate is changing at an unprecedented rate and threatening the service life of civil structures and infrastructural systems [1,2,3]. Therefore, the design of structures and the employment of building materials must take into account the changing background scenarios for engineering, such as escalating thunderstorms and waterlogging weather, increased loads, and electrically conductive surroundings. Glass fiber-reinforced polymers (GFRP) are light-weight and have excellent strength, high corrosion resistance, insulation, and durability, and could represent a relatively ideal partial or complete substitute for steel, could function in coordination with concrete, and could strengthen structures and resist corrosion and conduction, increasing the possibility of adapting structures to changes in harsh weather environments [4,5,6,7,8,9,10,11,12].

In engineering applications, GFRP often exists in the form of tubes and bars in concrete composites. Although constrained by GFRP tubes, concrete members can exhibit excellent strength, stiffness, and structural stability under tension and compression conditions [13,14,15,16,17,18]; without GFRP tubes, the performance of the reinforced members can be significantly affected by the behavior of the bars. When GFRP is combined with concrete in the form of bars, the mechanical characteristics of anisotropy, linear elasticity, and brittleness of GFRP cause their bonding behavior to differ significantly from that of steel-reinforced members [19,20,21], and the former introduces more complexity into the force transfer mechanism [22]. Scholars have conducted many experimental studies to clarify the bonding properties of GFRP bars and concrete and achieved fruitful results. Gudonis et al. [23] compared the test results of GFRP bars and steel bars in beam and pull-out tests and found that the bond strength measured by the pull-out test was better than that measured in the beam test under the same conditions [23,24,25]. In addition, Li et al. [26,27,28,29] developed a series of monitoring methods to nondestructively evaluate interfacial adhesion and reported in detail the effects of multiple variables on interfacial adhesion. Achillides et al. [30] found that the bonding mechanism between GFRP bars and concrete included chemical, frictional, and mechanical bonding. Okelo et al. [31] studied the bonding behavior of FRP bars and steel bars for different concrete compressive strengths and found that as the strength of concrete increased, the grip of the concrete on the GFRP bars and the ability to prevent the GFRP bars from being pulled out increased. The bond strength was proportional to the compressive strength of the concrete [32,33,34,35]. Tang et al. [36] noted that the growth rate of the bond strength slowed with increasing concrete strength. Henin et al. [37] studied the local bond–slip behavior of helically wrapped, sand-coated, deformed GFRP bars with four diameters, namely 10, 16, 19, and 25 mm. The bond strength was greater for bars with smaller diameters, and Yoo et al. [38,39,40] reported a similar relationship. Okelo and Robert [31] and Pecce et al. [41] studied the failure modes of GFRP bars and concrete specimens with different concrete strengths and bond lengths. They observed that for specimens with shorter embedded lengths and lower compressive strengths, the debonding failure mode was usually pull-out failure, while splitting failure was more likely at greater concrete strengths or bond lengths. Masmoudi et al. [42] found that GFRP-reinforced concrete members maintained good bond strength after being exposed to environments at 40 and 60 °C for 8 months. Fursa et al. [43] found that cyclical freezing and thawing led to the significant weakening of the bond strength between steel bars and concrete but adding a certain amount of fiber to concrete provided good resistance and control of the formation, development, and diffusion of interfacial cracks in concrete. Kim et al. [44] studied GFRP bar–concrete specimens containing different fibers (steel, polypropylene (PP), and polyvinyl alcohol (PVA)) and found that all three types of fibers effectively improved the initial toughness and bond strength. In addition, researchers [31,37,45,46,47] found that the surface treatments (sand coating, deformation, grooves) of GFRP bars significantly improved interfacial bonding capacity with concrete and enhanced ductility and bonding.

In addition, many scholars conducted research on predicting the bond behavior of GFRP bars and concrete by establishing numerical models. Malvar [48] studied the influence of lateral constraints and transverse rib height on bonding behavior, but because the model did not analyze the bond–slip process in sections, the prediction of the descending stage of the slip was not ideal. Eligehausen et al. [49] proposed the Bertero–Popov–Eligehausen (BPE) model, according to the bond–slip relationship between steel bars and concrete and divided the slip phase into ascending, horizontal, descending, and residual segments. Cosenza et al. [50] proposed a revised and modified Bertero–Popov–Eligehausen (mBPE) model after repeated calibrations through experiments and removed the horizontal section of the model to make it applicable to the analysis of FRP-reinforced concrete structures. Cosenza et al. [51] also proposed a Cosenza–Manfredi–Realfonzo (CMR) model for describing in detail the ascending section of FRP bar–concrete bond–slip. However, due to the lack of records of the sliding behavior of components during the entire stress process, it was difficult to analyze the ductility of the member. To better present the whole process of sliding of ribbed GFRP bars, Hao et al. [52] divided the bonding and sliding process between GFRP bars and concrete into four stages for detailed analysis. Gao et al. [53] proposed a continuous curve model, including the ascending and descending stages, according to the key points in the sliding process.

The application of GFRP bars in concrete structures has been widely established in a large number of studies. However, the influence of the surface deformation characteristics of GFRP bars on bonding performance is not yet fully understood. Therefore, a central pull-out test was designed according to the ACI440.3R-04 [54] standard. The influences of the deformation coefficients of the GFRP bars, concrete strengths, and diameters on the bond strengths and failure modes were analyzed. Based on the existing bond–slip constitutive model of GFRP bars and the test results, a curve prediction model, considering that bonding behavior engaged entire stages, was proposed. Additionally, a “four-fold model” involving representative stress was estimated.

## 2. Methodology

### 2.1. Materials

The concrete strength grades used in this test were C30 and C40 by BS EN12390 [55], and their mix proportions and properties are shown in Table 1. The measured compressive strengths of C30 and C40 after 28 days of standard curing were 31.40 MPa and 41.25 MPa, respectively. The GFRP bars used in this study were divided into three types according to their deformation coefficients: S1, S2, and S3. These GFRP bars were provided by Shandong Safety Industrial Co., Ltd. (Taian, China). The matrix resin was 25% epoxy resin by volume, and the reinforcing material was 75% glass fiber. The longitudinal tensile properties of the GFRP bars were tested according to specification [54], and the elastic modulus and tensile strength of three types of bars are shown in Table 2. Although the production process was similar, deformation characteristics were able to bring about small changes in the tensile elastic modulus and the tensile strength of the reinforcements. The variation range of the tensile elastic modulus for 14 mm was within 5%, and the variation range of the tensile strength for 14 mm was within 15%. For the S2 groups, the changes in diameter had little effect on the tensile elastic modulus, but the tensile strength decreased with the increase in diameter, as shown in Table 2.

### 2.2. Test Design

The deformation coefficient, steel bar diameter, and concrete strength grade were used as parameters to design and study the bond–slip behavior between deformed GFRP bars and concrete. The test numbers and design parameters are shown in Table 3, and the preparation elements of the specimens are shown in Figure 1. According to ACI440.3R-04 [54], 14 sets of test pieces were designed, each test number corresponded to three specimens with the same parameters, and the cube size was 150 mm × 150 mm × 150 mm. Figure 1a shows the center pull-test pieces of the concrete cube mold. The bonded length of the specimen was 5 times the diameter (5d) or 50 mm, whichever was greater. Figure 1b shows the anchoring elements of the GFRP bar of the specimen. Outside the specimen, the free end of the GFRP bar protruded 20 mm from the concrete surface for the measurement of the displacement of the free end. Inside the specimen, the no-bonding section was wrapped with a PVC sleeve, which was used to control the bonded length and reduce the impact of the extrusion stress generated in the concrete at the loading end during the loading process on the test results.

The deformation coefficient (*R_c_*) is a descriptive parameter used in this paper to analyze the influence of the distribution of the transverse ribs of GFRP bars on the bond performance. It characterizes the deformation characteristics of GFRP bars and is numerically equal to the area of adjacent interrib concrete (*A_ci_*), divided by the sum of the area of interrib concrete (*A_ci_*) and the area of individual cross-rib surface projections (*A_ur_*), where the sum of the area of *A_c_*_i_ and the area of *A_ur_* is equal to the product of rib spacing and rib height. GFRP bars obtain different deformation coefficient values according to Equation (1). Figure 2a shows three kinds of GFRP bars with different deformation coefficients, and their common deformation characteristics are shown in Figure 2b.
(1)$$R_{c}=\frac{A_{ci}}{A_{ci}+A_{ur}}$$

This test was performed on a 600 kN hydraulic servo testing machine, and the loading device is shown in Figure 3. The specimen prepared by the mold shown in Figure 1a was placed in the hanging basket, as shown in Figure 3, and an alignment hole was opened at the center of the lower plate of the hanging basket to ensure that the load was always applied along the axis of the reinforcement during the whole stress process to avoid the premature formation of cracks in the concrete specimen due to eccentric stress. The loading mode was controlled by displacement, and the loading rate was 1.2 mm/min. A displacement dial indicator was set on the free end of the GFRP bar to measure the slip level of the free end of the GFRP bar bond specimen, and two dial indicators were set equidistant on both sides of the bar and the unbonded section of concrete to measure the slip amount of the loading end of the bond specimen. The data were collected using the calculation method of Equations (2)–(4) [24], and the bond stress (*τ*) and its corresponding slip (*s*) of all specimens were obtained during the drawing period.
(2)$$\tau =\frac{F}{\pi dl_{a}}$$
(3)$$S=S_{a}-S_{b}$$
(4)$$S_{b}=\frac{FL}{EA}\;$$
where *τ* is the average bond stress between the GFRP bars and concrete, *F* is the pull-out load, *l_a_* is the bonding length between the GFRP bars and concrete, *S* is the slippage of the GFRP bars at the loaded end, *S_a_* is the measured value of the indicator at the loaded end, *S_b_* is the theoretical elastic total elongation of the GFRP bar in the loaded section, *L* is the initial length of the GFRP bar in the loaded section, *E* is the elastic modulus of the GFRP bar, and *A* is the cross-sectional area of the GFRP bar.

## 3. Results

The bond–slip curves of each group of specimens obtained according to Equations (2)–(4) are shown in Figure 4a–f, showing the S1/S2/S3 class of GFRP rebar in C30 and C40 concrete, respectively. The test results showed that the specimens with unique deformation coefficients presented distinguishing curve characteristics. A typical bond–slip curve consisted of four stages: linear rising, nonlinear rising, falling, and residual stages. At the initial stage of loading, the bond stress and slip between the GFRP bars and concrete were small, and the *τ*–*s* curve of the specimen rose nearly linearly. With continued loading, the *τ*–*s* curve entered a nonlinear rising stage and the slope of the curve decreased with increasing stress until it reached the peak bond stress (bond strength). Then, the curve began to enter the descending phase, and the bond stress decreased rapidly with increasing slip. The residual stage presents two forms: one is where the bond stress maintains a relatively stable value with increasing slip deformation, such as the S1 and S2 GFRP bars, shown in Figure 4a,b,d,e, and the other is where the stress decreases with the amount of slip in a wave-like manner, such as the C30S3 specimens, shown in Figure 4c. However, the ductility of C40S3, as shown in Figure 4f, was significantly insufficient. It shows the peak bonding stress of C40S3 specimens were not fully exerted due to their split failure.

In the experiment, 93% of the specimens failed due to the pull-out of the GFRP bars. In this case, the GFRP bars were pulled out from the concrete, there were practically no obvious scratches on the surface of the GFRP bars, the surface of the concrete test block was intact, and no radial cracks were observed; these observations indicated typical pull-out failure (P). For approximately 7% of the total number of specimens, splitting failure (S) was observed where the concrete specimen split directly in the middle and formed two similar halves. The specific failure modes and the test results of the bond strength are shown in Table 3.

According to Table 3, while the deformation coefficient increased from 39% (S1) to 44% (S2) and 53% (S3), the bond strengths for C30S2 and C30S3 increased by factors of 6.3 and 8.4 relative to that of C30S1, and by 5.3 times and 8.5 times, respectively, for C40S2 and C40S3, relative to that of C40S1. Additionally, as the compressive strength of the concrete increased from 31 to 41 MPa, the bond strengths of the specimens increased by 0.20 to 0.44 times. While compared with the C30S3 specimen, as shown in Table 3, the C40S3 specimens had greater bond strengths of a factor of approximately 0.44. However, the brittleness of C40S3 was prominent, which makes its use inadvisable for engineering applications. It is obvious that with the increase in the rib deformation coefficient or the concrete strength grade, the bond strength increases. Additionally, the test results showed that the smaller the diameter of the GFRP bar, the stronger the bond strength, which exhibits a similar material condition of the deformation coefficient and the concrete strength. In the C30S2 specimen, the bond strength of the specimen decreased by approximately 5% on average per 2 mm increase in the diameter of the GFRP bar; however, there was no obvious monotonic relationship between the diameter and the ductility of the member, as shown in Figure 4b,e and Table 3.

## 4. Engineering Characteristics

As shown above (Section 3), specimens with different deformation coefficient exhibited distinct four-stage bond–slip curves. For engineering applications, the *τ*–*s* curve of the residual phase decreasing in a wave-like manner indicates that during the relative sliding between the GFRP bar and the concrete, the overall member retained its bonding strength to some extent. Meanwhile, members that encompassed the typical four-stage process usually avoided sudden failure occurrence and represented ample engineering potential. Experimentally, there was no obvious wave-like development trend of the other specimens in the falling stage except C30S3, and their bond stress values generally decreased monotonically with increasing slippage. The reserved strength in these specimens was insufficient, and signs of ductility before failure were not obvious, meaning that it would be inadvisable to directly employ this process in practical engineering. In reality, the deformation capability and the energy consumption of resisting deformation prior to structural damage should be fully considered in structural designs.

The principle of the energy dissipation in composite components to prevent bond–slip failure could be imitated to some extent to that of the single material components against structural failure. In terms of the bonding component, the failure is triggered by external load and the member attempts to resist structural damage by releasing the energy generated by the slip process. However, thanks to the irreversibility of the slip deformation, the energy generated by the bond–slip of component differs from the energy consumed by single material component, as they act against deformation until failure. To distinguish from the local recoverability in single material components, the ability to resist deformation in bonded components via energy dissipation in their linear elastic stage was defined as the imaginary resilience (*IR*), which is herein numerically equal to the integral area of the linear segment of the *τ*–*s* curve. The ability to resist failure through energy dissipation both in the nonlinear elastic and in plastic stages is defined as the imaginary toughness (*IT*), which is numerically equal to the rest of the integral area after the removal of the linear segment of the *τ*–*s* curve. The entire energy consumption (*EC*) is numerically equal to the total area enclosed by the *τ*–*s* curve. The finite slip between the final slip (*s_f_*) and peak slip (*s_u_*) is used to characterize the ductility (*D*) of the component, and the specific expression is shown in Equations (5)–(7), where *s_p_* and *s_f_* are the amount of slip at the end of the linear stage and the amount of slip at bond failure, respectively.
(5)$$IR=\int_{0}^{s_{f}}\tau (s)ds$$
(6)$$IT=\int_{s_{p}}^{s_{f}}\tau (s)ds$$
(7)EC=IR+IT

As a result, in Table 3, the imaginary resilience of GFRP bars and concrete composite members was relatively small, accounting for 1–5% of the corresponding imaginary toughness. This indicates the capacity to resist bonding failure mainly depended on the imaginary toughness. Shown obviously in Table 3, the deformation coefficient of the GFRP bars had the most significant effect on improving the bond strength, ductility, and energy dissipation of composite members. However, C40S3 exhibited much less ductility than that of C30S3 despite engaging in greater peak stress. Moreover, the effect of the diameter on the energy consumption of the elements was more likely to be reflected in larger concrete strength, as shown in Table 3.

The comparison with the bond–slip behavior of commonly used steel bars [56,57,58], as illustrated in Figure 5, shows that typical reinforced concrete members went through linear rising, nonlinear rising, falling, and residual stages before bonding failure. The members reached peak bonding stress faster than the GFRP bars, followed by a decrease after the peak point until bonding failure. Table 4 shows the quantitative analysis results of GFRP bars and steel bars in terms of bonding performance. The results show that compared with common ribbed steel bars in C30 concrete, S3 GFRP with apparent surface deformation not only reveals similar bond strengths as steel bars, about 74–79%, but the total energy dissipation and ductility of S3C30 members are 2–2.5 times higher relative to steel members. These excellent mechanical properties provide a positive theoretical basis for GFRP to replace steel in such structural designs. When the concrete strength increased to C40, the energy dissipation of the steel members significantly improved and was approximately two times greater than that of C30. However, for the S3 GFRP, the increase in concrete strength changed the ductility, and subsequent failures, of the composite members from ductile to brittle. Thus, it is impracticable to seek to increase the bond strength by blindly increasing the strength of concrete. In engineering applications, much more attention should be paid to the joint effect on both of the bar deformation coefficient and concrete strength in order to achieve a composite design with a balance of structural strength and engineering performance.

## 5. Bond–Slip Model Prediction

### 5.1. Calibration of Test Results with Existing Model

In Section 3, we discussed the four-stage curves of the S1, S2, and S3 specimens. In reality, the previous descriptive function of the bond–slip relationship contained many undetermined factors pertaining to variable parameters. As shown in Figure 6, five types of bond–slip models [48,50,51,52,53] were compared with the test results for the S1, S2, and S3 GFRP bars (d = 14 mm). An analysis showed that the bond–slip model proposed by Malvar et al. [48] failed to separate the slip stage, which resulted in a fitting effect in the falling section that was not ideal. The mBPE model [50] has a good prediction effect on GFRP bars (S1,S2) with a small deformation coefficient, but it is difficult to achieve the expected prediction effect if the residual stage of wave attenuation occurs. It is obviously conservative to estimate the sliding descent stage as a linear change [50,52], while Gao et al. [53] were more objective in predicting the descending stage. When GFRP bars show wave-like attenuation in the residual stage due to a large deformation coefficient, the prediction effect of Hao et al. [52] is better. In order to better display the influence of deformation coefficient on the bond constitutive law of GFRP, the following suggestions are proposed In order to predict the bond–slip of GFRP bars with different deformation coefficients.

Before the residual stage (0 ≤ *s* < *s_r_*):(8)$$\tau =\frac{s}{as^{2}+bs+c}$$

Residual stage (*s* ≥ *s_r_*):(9)$$\tau =\tau_{r}-\gamma [e^{-\xi \omega (s-s_{r})}\cos\omega (s-s_{r})-1]+\rho [e^{-\xi \omega (s-s_{r})}-1]$$
where *a*, *b*, *c*, *γ*, *ξ*, *ω*, and *ρ* are all curve fitting parameters and *τ_r_* is the residual bond stress.

In terms of the two divided stages, as shown in Figure 6, the residual stage described by Hao et al. [52] is recommended in order to effectively assesses the complex characteristics of GFRP bars in the residual stage. Additionally, the comparison results show that the entire prediction model results are extremely close to those of the actual working conditions, as shown in Table 4 and Figure 7.

### 5.2. Simplified Constitutive Model

In fact, it is often very important for engineering applications and structural designs to grasp the essential characteristics and engineering characteristics of bond–slip relationships. Therefore, in this study, we employ a simplified model for GFRP bonding performance in order to designate *τ_p_*, the linear elastic peak stress; *τ_u_*, the nonlinear elastic peak stress; and *τ_r_*, the stress that tends to be stable in the residual stage, or the end point of the descending stage if wave attenuation occurs. According to these representative values of bond stress and the corresponding slip, a simplified linear bond–slip model can be established, as Figure 8 shows. The typical ideal linear model should be a four-fold model. Additionally, a new bond–slip constitutive model can be proposed, as shown in Equations (10)–(13):

Linear rising stage (0 ≤ *s* ≤ *s_p_*):(10)$$\tau =\frac{\tau_{p}}{s_{p}}s$$

Nonlinear rising stage (*s_p_* ≤ *s* ≤ *s_u_*):(11)$$\tau =\frac{\tau_{u}-\tau_{p}}{s_{u}-s_{p}}(s-s_{p})+\tau_{p}$$

Falling stage (*s_u_* ≤ *s* ≤ *s_r_*):(12)$$\tau =\frac{\tau_{r}-\tau_{u}}{s_{r}-s_{u}}(s-s_{u})+\tau_{u}$$

Residual stage (*s* ≥ *s_r_*):(13)$$\tau =\tau_{r}$$

The comparison of the simplified model with the test curve are shown in Table 4. The polygonal area in the simplified model was directly calculated according to the representative values in order to analyze the energy dissipation capacity of the component (*EC^s^*). As shown in Figure 9, the increased areas and decreased areas reveal the differences in the calculated results between the simplified model and the test curve. In reality, the *EC* predicted by the simplified model is often slightly lower than the actual *EC*, at about 83–99% of the actual *EC*, as shown in Table 4. According to the results of the analysis, the four-fold model, which, when slightly conservative, is able to provide a better grasp of the original constitutive characteristics and engineering performance.

## 6. Conclusions

In this paper, a central pull-out test of GFRP bar and concrete specimens was designed and the results were analyzed in order to investigate the effects from the deformation coefficient, concrete strength, and diameter of GFRP bars on the bond strength, as well as the failure mode and bond–slip constitutive formulas. The applicability of the bond–slip constitutive models was verified, and the conclusions are drawn as follows:While increasing the deformation coefficient from 39% (S1) to 44% (S2) and 53% (S3), the bond strength for C30S2 and C30S3 increased by 630% and 840%, respectively relative to that of C30S1, and by 530% and 850%, respectively, for C40S2 and C40S3, relative to that of C40S1. Additionally, when increasing the compressive strength of the concrete from 31 to 41 MPa, the bond strengths of the specimens increased by 20%–44%. Meanwhile, the bond strength of the specimen decreased by approximately 5% on average per 2 mm when increasing the diameter of the GFRP bars.While both the deformation coefficient and concrete strength of the GFRP bars were increased, the bond failure mode of the composite member will be more likely changed from a typical pull-out failure to a concrete splitting failure. In engineering applications, much more attention should be paid to the joint effect on both the bar deformation coefficients and concrete strengths, in order to achieve a composite design with a balance of structural strength and engineering performance.Members with larger deformation coefficients and moderate concrete grades generally have excellent mechanical and engineering properties. The S3 GFRP bar was recommended for use in concrete C30 as an alternative to steel bars thanks to its superior performance. The total energy dissipation and ductility of C30S3 members are 200% to 250% relative to that of steel members.When compared with the existing bond and slip constitutive models, the proposed curve prediction model can better exhibit the unique influence of deformation coefficients on bonding behavior and the engineering performance of GFRP bars. The four-fold model, which is slightly conservative at approximately 83–99% of the actual energy consumption, is able to provide a better balanced design with original constitutive characteristics and engineering performance.

## Figures and Tables

**Figure 1 polymers-15-02604-f001:**
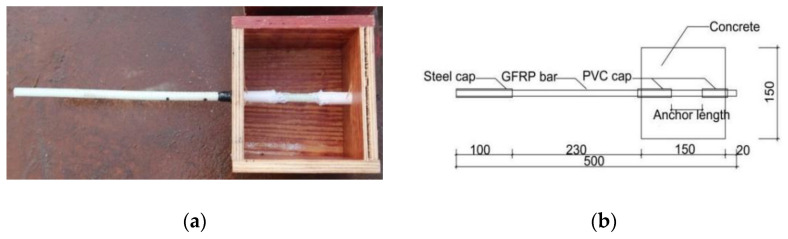
Specimen preparation: (**a**) Cube mold for Concrete and (**b**) Anchoring and casing of GFRP bar.

**Figure 2 polymers-15-02604-f002:**
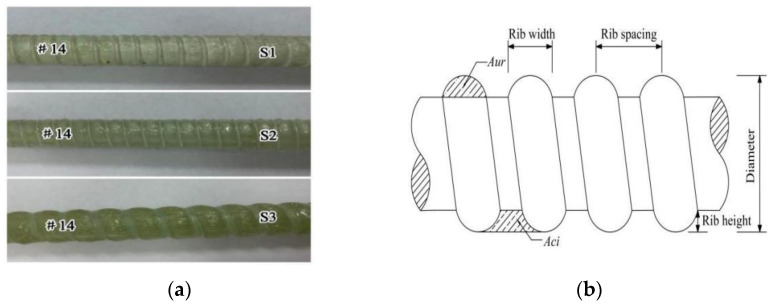
Surface deformations of GFRP bars: (**a**) three surface types of GFRP bars and (**b**) surface deformation schema of GFRP bars.

**Figure 3 polymers-15-02604-f003:**
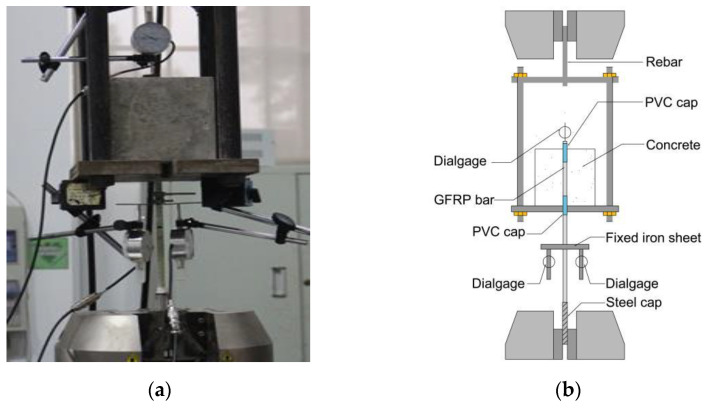
Loading device: (**a**) complete loading device and (**b**) components of loading device.

**Figure 4 polymers-15-02604-f004:**
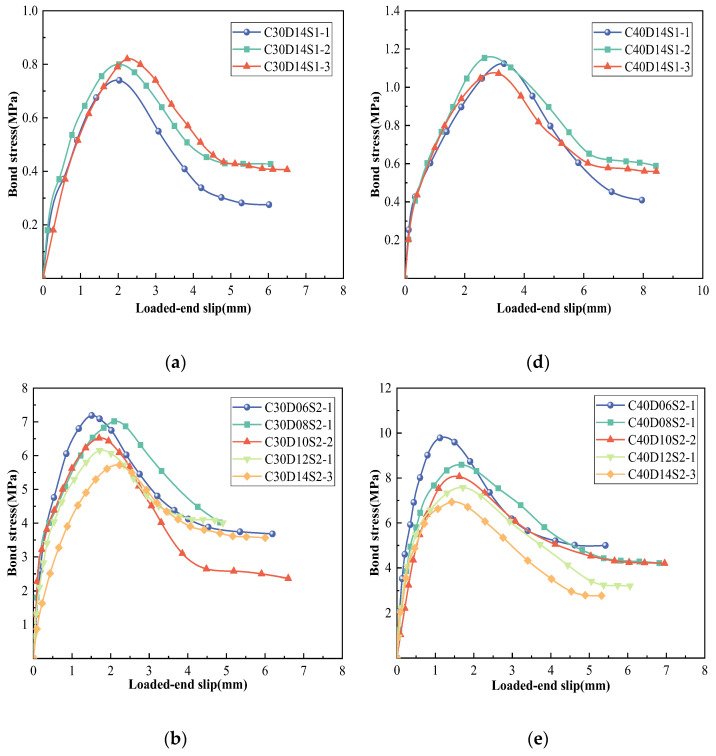
Bond–slip curves of specimens: (**a**) C30D14S1, (**b**) C30D06–14S2, (**c**) C30D14S3, (**d**) C40D14S1, (**e**) C40D06–14S2, and (**f**) C40D14S3.

**Figure 5 polymers-15-02604-f005:**
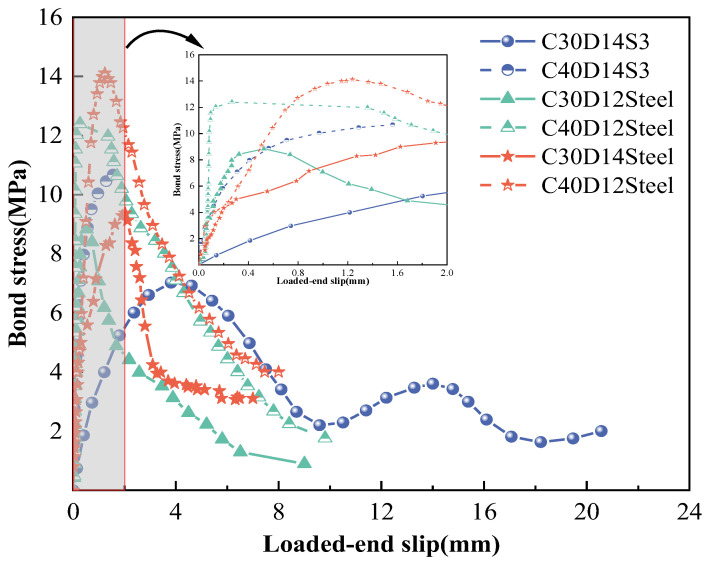
Comparison of bond–slip stages for GFRP bars and steel [56,57,58].

**Figure 6 polymers-15-02604-f006:**
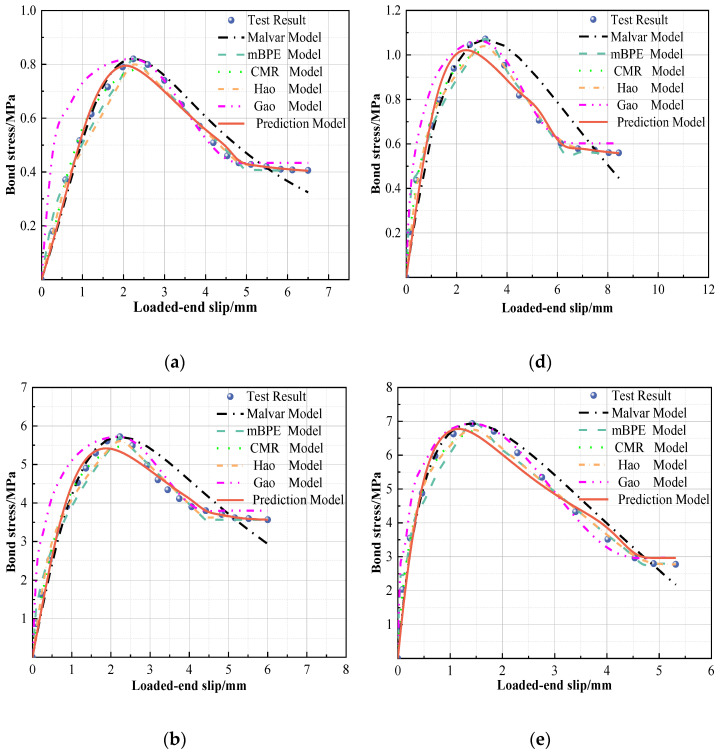
Test and model calibration results: (**a**) C30D14S1, (**b**) C30D14S2, (**c**) C30D14S3, (**d**) C40D14S1, (**e**) C40D14S2, and (**f**) C40D14S3 [48,50,51,52,53].

**Figure 7 polymers-15-02604-f007:**
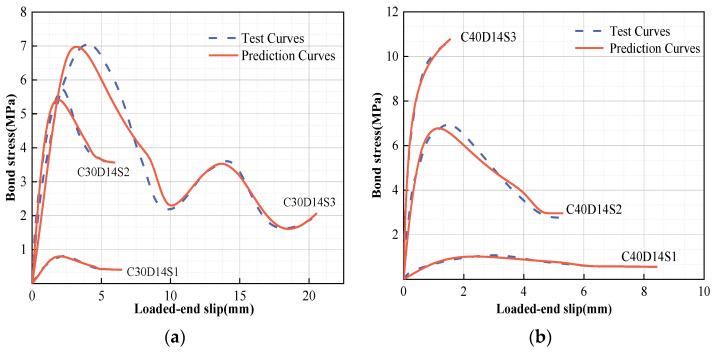
Proposed bond–slip models of GFRP bars: (**a**) C30D14S1–S3 and (**b**) C40D14S1–S3.

**Figure 8 polymers-15-02604-f008:**
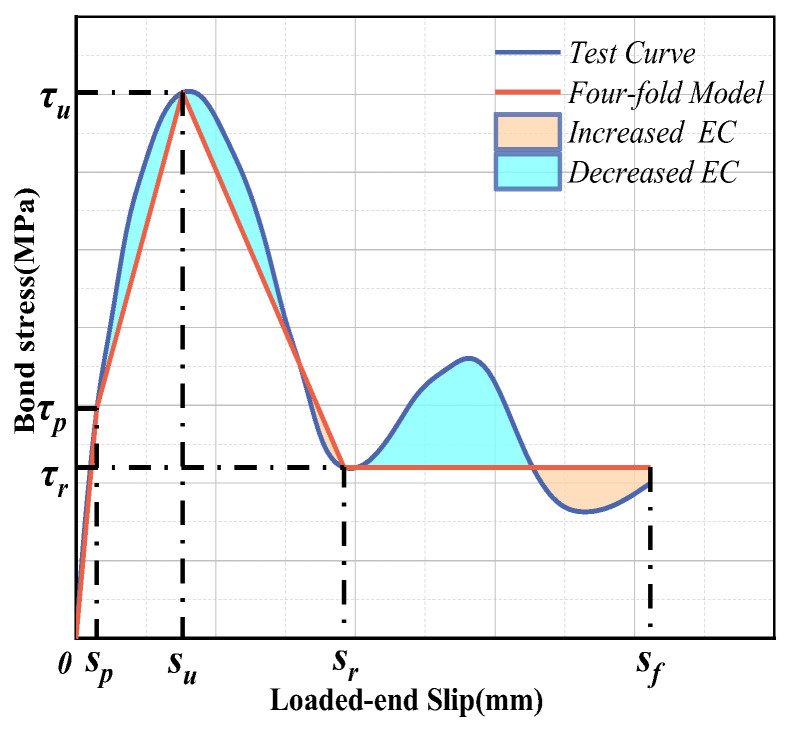
Four-fold model of GFRP bars.

**Figure 9 polymers-15-02604-f009:**
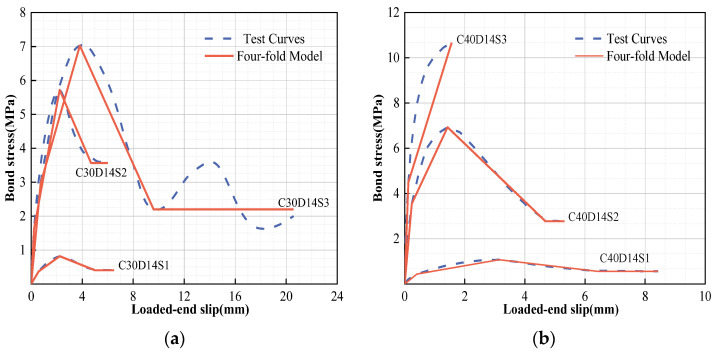
Comparison of in curve models and four-fold models for GFRP bond–slip performance: (**a**) C30D14S1–S3 and (**b**) C40D14S1–S3.

**Table 1 polymers-15-02604-t001:** Mix proportions and properties of concrete.

Concrete Type	Water (%)	Cement (%)	Fine Aggregate (%)	Coarse Aggregate (%)	Compressive Strength (MPa)
C30	7.7	13.5	27.6	51.2	31.40
C40	7.6	17.1	23.3	52.0	41.25

**Table 2 polymers-15-02604-t002:** Material properties of GFRP bars (average ± standard deviation).

GFRP Bar Type	Diameter (mm)	Epoxy Resin (%)	Glass Fibers (%)	Elastic Modulus (GPa)	Tensile Strength (MPa)
S1	14	25	75	50.00 ± 0.20	1013.54 ± 49.97
S2	6	25	75	46.10 ± 0.11	1264.76 ± 50.56
	8	25	75	47.16 ± 0.36	1149.10 ± 45.90
	10	25	75	46.84 ± 0.34	1101.73 ± 36.07
	12	25	75	48.21 ± 0.19	1096.27 ± 28.53
	14	25	75	50.71 ± 0.51	945.64 ± 18.53
S3	14	25	75	47.81 ± 0.20	860.00 ± 39.99

**Table 3 polymers-15-02604-t003:** Matrix of design variables and summary of test results (average ± standard deviation).

Test Specimen	Compressive Strength (MPa)	Diameter (mm)	Bar Type	Rib Height (mm)	Rib Spacing (mm)	Rib Width (mm)	*Rc* (%)	Anchor Length (mm)	*τ_u_ *(MPa)	Failure Mode	*D*	*IR*	*IT*	*EC*
C30D14S1	31.40	14	S1	0.10	5.00	4.60	38.64	70	0.79 ± 0.03	P	4.10	0.11	3.40	3.51
C30D06S2		6	S2	0.80	8.00	6.60	44.36	50	7.34 ± 0.29	P	4.68	0.23	29.71	29.94
C30D08S2		8						50	6.97 ± 0.16	P	3.59	1.01	24.82	25.83
C30D10S2		10						50	6.69 ± 0.34	P	4.90	0.44	25.79	26.23
C30D12S2		12						60	6.12 ± 0.20	P	3.19	0.20	22.76	22.96
C30D14S2		14						70	5.78 ± 0.18	P	3.77	0.21	24.59	24.80
C30D14S3		14	S3	1.50	10.00	6.80	52.95	70	7.41 ± 0.27	P	17.76	1.19	72.98	74.17
C40D14S1	41.25	14	S1	0.10	5.00	4.60	38.64	70	1.12 ± 0.03	P	5.23	0.13	6.09	6.22
C40D06S2		6	S2	0.80	8.00	6.60	44.36	50	9.75 ± 0.31	P	4.30	0.26	35.80	36.06
C40D08S2		8						50	8.66 ± 0.23	P	5.14	0.54	40.17	40.71
C40D10S2		10						50	8.04 ± 0.27	P	5.32	0.96	37.09	38.05
C40D12S2		12						60	7.62 ± 0.23	P	4.34	0.62	31.13	31.75
C40D14S2		14						70	7.03 ± 0.18	P	3.89	0.52	24.76	25.28
C40D14S3		14	S3	1.50	10.00	6.80	52.95	70	10.69 ± 0.18	S	0.00	0.34	13.21	13.55

**Table 4 polymers-15-02604-t004:** Bar–concrete bond–slip properties and engineering characteristics (diameters 14 mm).

Rebar Type	Concrete Type	D	Representative Stress			Test Results			Four-Fold Model			Curve Model	
			*τ_p_ *(MPa)	*τ_u_ *(MPa)	*τ_r_ *(MPa)	*IR*	*IT*	*EC*	*IR^s^*	*IT^s^*	*EC^s^*	*R* ^2^	*EC^f^*/*EC*
S1 class GFRP bar	C30	6.19	0.37	0.82	0.41	0.11	3.40	3.51	0.11	3.28	3.39	0.9826	0.9930
	C40	8.27	0.47	1.07	0.56	0.13	6.09	6.22	0.09	5.87	5.96	0.9680	0.9886
S2 class GFRP bar	C30	6.00	1.63	5.72	3.56	0.21	24.59	24.80	0.18	23.47	23.65	0.9825	0.9952
	C40	5.32	3.55	6.93	2.77	0.52	24.76	25.28	0.42	23.81	24.23	0.9722	0.9965
S3 class GFRP bar	C30	21.31	2.96	7.02	2.00	1.19	72.98	74.17	1.10	66.16	67.26	0.9552	0.9908
	C40	1.38	4.50	10.67	–	0.34	13.21	13.55	0.27	10.93	11.20	0.9976	0.9973
Steel bar [56]	C30	9.01	7.97	8.83	0.90	1.16	27.25	28.41	1.05	26.60	27.75	–	–
	C40	9.81	11.54	12.40	1.76	0.27	61.55	61.82	0.56	60.83	61.39	–	–
Steel bar [57]	C30	7.00	4.71	9.38	3.11	0.77	34.13	34.90	0.63	31.96	32.59	–	–
Steel bar [58]	C40	8.00	4.00	14.11	1.76	0.34	61.45	61.79	0.25	60.70	60.95	–	–

Notes: *τ_p_*, *τ_u_*, and *τ_r_* are the representative stresses; *IR*, *IT*, and *EC* are the actual energy consumption parameters; *IR^s^*, *IT^s^*, and *EC^s^* are the energy consumption parameters under the simplified model; *R^2^* is the coefficient of determination; *EC^f^* is the energy consumption under the predicted flexural model; “–” pertaining to GFRP bar means that the value was not obtained in the test due to specimen failure. Other instances of “–” indicate that the reinforced concrete members contained distinguishable intrinsic relationships [59,60,61].

## Data Availability

The authors confirm that the data supporting the findings of this study are available within the article.

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
