# Peer review of "Bonding Performance of Glass Fiber-Reinforced Polymer Bars under the Influence of Deformation Characteristics"

_polymers, 2023, doi:10.3390/polym15122604_

Round 1

Reviewer 1 Report

Paper title: Bonding performance of GFRP bars under the influence of deformation characteristics

·         The paper assesses the effects of GFRP bars with various diameters on bonding properties of GFRP-reinforced concrete. Pull-out tests were conducted to investigate different failure modes of the proposed materials.

·         The manuscript presents interesting topic with clear structure and appropriate experimental/theoretical-based methods. However, the abstract and introduction need to emphasize justification of selected methods as well as possible implications of their study.

·         The two sections also have some irrelevant and unfocused paragraphs, for instance, in lines 12-13 and 37-40, how can the aforementioned properties effectively mitigate adverse effects of climate change?

·         In table 1, mix proportions and properties of concrete should be separated with the properties of GFRP bars.

Author Response

Dear Reviewer:

We would like to thank you very much for your valuable comments and good suggestions that greatly helped to improve our manuscript (ID:polymers-2365224). We have considered the comments very carefully and have revised the paper accordingly. The reviewer comments are laid out below in italicized font and specific concerns have been numbered. Our response is given in normal font and changes/additions to the manuscript are given in the yellow text. Revision notes, point-to-point, are given as follows:

Comment 1 The abstract and introduction need to emphasize justification of selected methods as well as possible implications of their study.

Response 1 In the abstract and introduction sections, we have emphasized the justification of the selected methods and made the implications of the study more refined. The modifications of the abstract and introduction are shown in lines 10–12, 15, and 30-39 and 104, respectively, as given with yellow highlight. Thank you very much.

Comment 2 The two sections also have some irrelevant and unfocused paragraphs, for instance, in lines 1213 and 3740, how can the aforementioned properties effectively mitigate adverse effects of climate change?

Response 2 We have re-written the two section parts according to the your suggestion. That is the design of the structure and the employment of building material must take into account the updating engineering scene, such as escalating thunderstorm and waterlogged weather, increased loads, and electrically conductive surrounding, etc. Glass fiber-reinforced polymer (GFRP) has light weight, excellent strength, and high corrosion resistance, insulation and durability, as a relatively idea substitution partially or completely for steel, work coordinated with concrete, could strengthen the structures, resist corrosion, and resist conduction, increasing the possibility of adapting structure to changes in harsh weather environments. The modifications of the abstract and introduction can be found in lines 10–12 and 30–39, page 1 of the revised manuscript. Thank you very much.

Comment 3 In table 1, mix proportions and properties of concrete should be separated with the properties of GFRP bars.

Response 3 We have separated the properties information for concrete and GFRP bars, employing Table 1 and Table 2, respectively. In the updated revised manuscript, The exact location of the change can be found in lines 127–128, page 3 of the revised manuscript. Thank you very much.

Other Changes:

  1. Updated the line numbers of the manuscript.

  1. Line 120, the statements of “tensile strength and elastic modulus”were corrected as “elastic modulus and tensile strength”.

  1. Line 123,124, added ”for 14mm”.
  2. Line 124, the statements of “10%”were corrected as “15%”.

  1. Line 200, we corrected the data as “3; 8.4; 5.3; 8.5”based on calculation results.

  1. The numbers of tables have updated since “Table 2 “to “Table 4”.

  1. Line 266, the statements of “GFRP”were corrected as “steel”.

  1. Line 348, “Though the energy dissipation of theC40S3 significantly improved, approximately 2 times greater than that of C30S3, the increase in concrete strength changed the ductility of the member” was deleted, based on the analysis conclusion.

  1. Literatures[4]-[12] have been updated according to the modification of introduction.

  1. Explanations for Questions are provided in the responses, butnot highlight in the article.

Thank you for the time and so much again for all your precious advice and warm work earnestly. We sincerely hope the correction will meet with approval. 

Best Regards,

Fang Xie

Reviewer 2 Report

This paper investigates the shape of GFRP and the bond strength according to the compressive strength of concrete. It presents an appropriate experimental program and model confirmation. However, it is judged that some questions and corrections will be needed, as shown below.

1. It is necessary to separate the contents of Table 1 (Concrete, CFRP bar)

2. Express the error of GFRP Elastic modulus and Tensile strength in Table 1 in the same unit (% -> GPa and MPa)

3. Need to adjust the position of Table 2 and Table 3

4. Present the number of specimens and standard deviation for the test results

5. The subheadings (b) and (e) of Figure 4 and the specimen number in the figure are different. It is necessary to change the subheading or separate the graph for each specimen.

6. There is no model that predicts the change in bond strength according to the diameter of S2, and it is unknown whether the tensile strength is the same regardless of the diameter.

It is necessary to review the thesis by revising the relevant part.

Author Response

Dear Reviewer:

We would like to thank you very much for your valuable comments and good suggestions that greatly helped to improve our manuscript (ID:polymers-2365224). We have considered the comments very carefully and have revised the paper accordingly. The reviewer comments are laid out below in italicized font and specific concerns have been numbered. Our response is given in normal font and changes/additions to the manuscript are given in the yellow text. Revision notes, point-to-point, are given as follows:

Comment 1 It is necessary to separate the contents of Table 1 (Concrete, GFRP bar).

Response 1 We have separated the properties information for concrete and GFRP bars, employing Table 1 and Table 2, respectively. In the updated revised manuscript, The exact location of the change can be found in lines 127–128, page 3 of the revised manuscript. Thank you very much.

Comment 2  Express the error of GFRP Elastic modulus and Tensile strength in Table 1 in the same unit (% -> GPa and MPa)

Response 2 We have re-edited this part according to the your advice. The modifications are shown in line 128, page 3 of the updated manuscript. Thank you very much.

Comment 3 Need to adjust the position of Table 2 and Table 3 

Response 3 The original Table 2 and Table 3 were converted to Table 3 and Table 4 according to your comments according to the article structure. And the modifications are shown in page 5 and page 6 of the revised manuscript. Thank you very much.

Comment 4 Present the number of specimens and standard deviation for the test results

Response 4 The results are the representative values in each group of specimens previously. Now the number of samples are clearly presented in the updated manuscript, and the changes can be found in lines 134–135, page 4. Meanwhile, the standard deviation of the test results presently can be found in Table 3 of the updated page 5. Thank you very much.

Comment 5 The subheadings (b) and (e) of Figure 4 and the specimen number in the figure are different. It is necessary to change the subheading or separate the graph for each specimen.

Response 5 We have corrected (b) and (e) of Figure 4, for C30 and C40 separated, shown in page 9 of the updated manuscript. Thank you very much.

Comment 6 There is no model that predicts the change in bond strength according to the diameter of S2, and it is unknown whether the tensile strength is the same regardless of the diameter.

Response 6 As our tests and studies shown, the bond strength of GFRP bar decreases with the increase of diameter. Because the course of the effect diameter delivered simply and monotonically, the model was not discussed further more here, and it has briefly described in lines 207-210 and illustrated in Fig. (b) and (e) of Figure 4. Meanwhile, Since tensile strength is not the most significant factor in this study, there is few extraordinarily detailed analysis was presented. We have presented the pertaining analysis in the manuscript, as shown in lines 124-126, page 3. Actually, based on the previously analysis in our research, the influence of the diameter on the tensile strength and shear strength practically are obvious, as literature [21] shown, that is the diameter has a significant effect on the tensile and shear strength of GFRP bars; when the diameter increases from 8mm to 16mm, the average tensile strength of the specimen decreases by about 15%; the shear strength decreases by about 10%. Thank you very much.

Other Changes:

  1. Updated the line numbers of the manuscript.

  1. Line 120, the statements of “tensile strength and elastic modulus”were corrected as “elastic modulus and tensile strength”.

  1. Line 123,124, added ”for 14mm”.

  1. Line 124, the statements of “10%”were corrected as “15%”.

  1. Line 200, we corrected the data as “3; 8.4; 5.3; 8.5”based on calculation results.

  1. The numbers of tables have updated since “Table 2 “to “Table 4”.

  1. Line 266, the statements of “GFRP”were corrected as “steel”.

  1. Line 348, “Though the energy dissipation of theC40S3 significantly improved, approximately 2 times greater than that of C30S3, the increase in concrete strength changed the ductility of the member” was deleted, based on the analysis conclusion.

  1. Literatures[4]-[12] have been updated according to the modification of introduction.

  1. Explanations for Questions are provided in the responses, butnot highlight in the article.

Thank you for the time and so much again for all your precious advice and warm work earnestly. We sincerely hope the correction will meet with approval. 

Best Regards,

Fang Xie

Reviewer 3 Report

1.      Abstract - “Engineers” Are sure about this?

2.      Line 115 – How were these results obtained? By the authors?

3.      Tables 1,2 and 3 – References? These tables were not evaluated in this study.

4.      Were samples inspected prior to testing? Are there any defects, such as voids, in the samples?

5.      How many tests were performed? Only three tests at each condition? Can the authors ensure the repeatability and reproducibility of the results?

6.      A table with mean values and respective standard deviations should be considered.

7.      The discussion should be done in terms of percentages, to make it clearer for the readers.

8.      What kind of defects can appear in these materials? Has the possible presence of defects been considered? Is there any influence on the performance of the materials?

9.      Poor discussion with open literature. The conclusions can be questionable. The results must be presented, discussed and justified.

Author Response

Dear Reviewer:

We would like to thank you very much for your valuable comments and good suggestions that greatly helped to improve our manuscript (ID:polymers-2365224). We have considered the comments very carefully and have revised the paper accordingly. The reviewer comments are laid out below in italicized font and specific concerns have been numbered. Our response is given in normal font and changes/additions to the manuscript are given in the yellow text. Revision notes, point-to-point, are given as follows:

Comment 1 Abstract - “Engineers” Are sure about this?

Response 1 We have re-written this part according to the your advice. And the modifications are shown in lines 10–12, page 1 of the revised manuscript. Thank you very much.

Comment 2 Line 115 – How were these results obtained? By the authors?

Response 2 The properties of GFRP bars and concrete are obtained by our actual measurements according to the specifications. The related information can be found in lines 112–126 and Table 1 and Table 2, page 3 of the updated manuscript. Thank you very much.

Comment 3 Tables 1,2 and 3 – References? These tables were not evaluated in this study. 

Response 3 The tables are the illustrations for our manuscript of research results. The properties information of concrete and GFRP, which presented in Table 1 and Table 2 are  information which examined and obtained by manufacturer before test, have been described in lines 112–126, page 3. The updated Table 3 and Table 4 are the information and pertaining results from this test, which are discussed in the lines 190–213, page 9 and lines 246–254, page 10 and in lines 259–273, page 11 and lines 316–324, page 14 of the revised manuscript. Thank you very much.

Comment 4 Were samples inspected prior to testing? Are there any defects, such as voids, in the samples?

Response 4 Before the test, each component was carefully checked and we made the concrete surface smooth and flat and the position of the GFRP bar set along the central axis in order to make sure there was no gap between the GFRP bar, PVC cap, and concrete in the block sample. Observations shown the results well meet the expectations. Thank you very much.

Comment 5 How many tests were performed? Only three tests at each condition? Can the authors ensure the repeatability and reproducibility of the results?

Response 5 In this study, we carried out a total of 14 sets of experiments, each with three duplicates. As shown in updated Table 3, the standard deviation of each group are within 5%, which has been presented in lines 134–135, page 4 and Table 3, page 5. Thank you very much.

Comment 6 A table with mean values and respective standard deviations should be considered.

Response 6 We have supplemented the standard deviation of the results in Table 3 as shown in Table 3, page 5 of the updated manuscript. Thank you very much.

Comment 7 The discussion should be done in terms of percentages, to make it clearer for the readers.

Response 7 We have revised the conclusion section in the updated manuscript according to your advice, as shown in lines 331–334, page 14 and lines 335–354, page 15 of the manuscript. Thank you very much.

Comment 8 What kind of defects can appear in these materials? Has the possible presence of defects been considered? Is there any influence on the performance of the materials?

Response 8 There are many kinds of GFRP bars, and the mechanical properties are greatly affected by factors such as different fiber volume content, matrix resin ratio and manufacturing process, which makes different batches of GFRP materials vary greatly. In this study, we employed the identical manufacture process for GFRP material, as shown in lines 116-118, and each test undergone a strict test condition control and completed in a certain period in order to evaluate them uniformly. Thank you very much.

Comment 9 Poor discussion with open literature. The conclusions can be questionable. The results must be presented, discussed and justified.

Response 9 Regarding the influence of concrete strength and diameter on bond performance, we introduced the research results of literature [31-36] and literature [37-40] respectively in the preface, as shown in lines 56–62 and lines 62–65 of updated page 2. Actually there is few relevant literature for the impact of deformation on the bonding performance. As our studies shown, the bond strength of GFRP bar decreases with the increase of diameter. Because the impact of the diameter is relatively obvious and the model of it is delivered monotonically, it was not discussed further more here, and it has briefly described in lines 207-210 and illustrated in Fig. (b) and (e) of Fig. 4. and the”four-fold model”is a novel accept in order to much simply and practically evaluate the bonding behavior of GFRP. Thank you very much.

Other Changes:

  1. Updated the line numbers of the manuscript.
  2. Line 120, the statements of “tensile strength and elastic modulus”were corrected as “elastic modulus and tensile strength”.
  3. Line 123,124, added ”for 14mm”.
  4. Line 124, the statements of “10%”were corrected as “15%”.
  5. Line 200, we corrected the data as “3; 8.4; 5.3; 8.5”based on calculation results.
  6. The numbers of tables have updated since “Table 2 “to “Table 4”.
  7. Line 266, the statements of “GFRP”were corrected as “steel”.
  8. Line348, “Though the energy dissipation of the C40S3 significantly improved, approximately 2 times greater than that of C30S3, the increase in concrete strength changed the ductility of the member” was deleted, based on the analysis conclusion.
  9. Literatures[4]-[12] have been updated according to the modification of introduction.
  10. Explanations for Questions are provided in the responses, andnot highlight in the article.

Appreciate your time and your precious advice.

Sincerely,

Fang Xie

Round 2

Reviewer 2 Report

I commend you for your efforts.

Reviewer 3 Report

The suggestions made by me were all accepted and the authors changed the paper.